# Ensembling geophysical models with Bayesian Neural Networks

**Ushnish Sengupta**[*]
University of Cambridge
Cambridge, UK
us271@cam.ac.uk

**Matt Amos**[*]
Lancaster University
Lancaster, UK
m.amos1@lancaster.ac.uk

**J. Scott Hosking**
British Antarctic Survey
Cambridge, UK

**Carl Edward Rasmussen**
University of Cambridge
Cambridge, UK

**Matthew P. Juniper**
University of Cambridge
Cambridge, UK

**Paul J. Young**
Lancaster University
Lancaster, UK

## Abstract

Ensembles of geophysical models improve prediction accuracy and express uncertainties. We develop a novel data-driven ensembling strategy for combining geophysical models using Bayesian Neural Networks, which infers spatiotemporally varying model weights and bias, while accounting for heteroscedastic uncertainties in the observations. This produces more accurate and uncertainty-aware predictions without sacrificing interpretability. Applied to the prediction of total column ozone from an ensemble of 15 chemistry-climate models, we find that the Bayesian neural network ensemble (BayNNE) outperforms existing methods for ensembling physical models, achieving a $49.4\%$ reduction in RMSE for temporal extrapolation, and a $67.4\%$ reduction in RMSE for polar data voids, compared to a weighted mean. Uncertainty is also well-characterized, with $91.9\%$ of the data points in our extrapolation validation dataset lying within 2 standard deviations and $98.9\%$ within 3 standard deviations.

## 1   Introduction

Climate models are the primary tool for predicting the evolution of Earth's uncertain climate and their output informs international policy [16]. Based on multi-scale physical and chemical processes they allow us to simulate conditions outside of the observational record, both spatially and temporally. Coordinated experiments with an ensemble of multiple models [11, 37] are typically used to increase the accuracy of prediction and to quantify predictive uncertainty, with studies often reporting predictions based on the ensemble average and uncertainty from the ensemble spread.

Such approaches assume that each separate climate model within the ensemble is independent and able to simulate the Earth system with equal skill, neither of which are true [12, 19]. Climate model skill is established through comparisons against a wide variety of remotely-sensed or in situ observations. Weighted measures of skill from these comparisons provide a more sophisticated method of combining an ensemble compared to simple averaging, and these weighted means are used to constrain ensemble predictions for single or multiple variables of interest [3, 20]. However, there are limitations of these approaches: they assume that historic climate model skills and behaviours can be translated to a future prediction; they rely on observations with the same spatiotemporal coverage as the models and cannot learn model weights for regions where data is sparse; they generally, do not account for the varying quality of observations upon which ensembles are weighted; and they do not account for spatially-varying skill in individual models.

---

[*]The first two authors contributed equally to this work. The order is arbitrary and determined by a coin flip.

Exploiting machine learning methods to analyse and process ensembles of climate models is an emerging area of research. Given the complexity and scale of the data involved, neural networks have obvious benefits. Key examples include emulating climate model ensembles [18], thus saving on high computational costs; identifying regional patterns of climate change [4]; and examining and quantifying differences between models' underlying representations of atmospheric physics and chemistry [30]. The comprehensive review of Reichstein et al. [35] provides further examples of how earth system science can – and has – benefited from neural networks and deep learning. Beyond deep learning, causal inference has provided a new way to analyse the skill of climate model ensembles [31], and newly proposed ensembling methods have improved climate model predictions [2, 26, 27, 40].

In this paper, we address the limitations of current ensembling methods and develop a method which provides more accurate and uncertainty aware predictions. Our approach combines geophysical models within a Bayesian ensembling framework, which assigns spatiotemporally varying weights to models and accounts for heteroscedastic aleatoric uncertainty in observational data, as well as epistemic uncertainty where data is unavailable. By fusing models, which codify our best physical knowledge, with observations, we can better interpolate and extrapolate geophysical data. This provides more accurate future predictions as well as spatiotemporally continuous and observationally constrained historic states. A key strength of our approach is the data model's interpretability, extending its use beyond its predictive capabilities to bring insight and understanding to the climate models.

The code and pretrained models accompanying this paper are hosted in a Github repository `https://github.com/Ushnish-Sengupta/Model-Ensembler`. The dataset of total column ozone observations [6] and chemistry-climate model predictions from 1980 to 2010 [28] are processed, combined and made available as a resource (`https://osf.io/ynax2/download`) for future studies in geophysical model ensembling.

## 2 Methods

### 2.1 Problem formulation

We assume that observations $y(\mathbf{x}, t)$ can be modelled as a sum of $n$ physical model predictions $M_i(\mathbf{x}, t)$ weighted by their respective weights $\alpha_i(\mathbf{x}, t)$, a bias term $\beta(\mathbf{x}, t)$ and a heteroscedastic aleatoric noise term $\sigma(\mathbf{x}, t)$.

$$y(\mathbf{x}, t) = \sum_{i=1}^{n} \alpha_i(\mathbf{x}, t) M_i(\mathbf{x}, t) + \beta(\mathbf{x}, t) + \sigma(\mathbf{x}, t) \tag{1}$$

Model weights form a partition of unity, i.e., $\alpha_i(\mathbf{x}, t) > 0$ and $\sum_{i=1}^{n} \alpha_i(\mathbf{x}, t) = 1 \, \forall \, \mathbf{x}, t$. The weights, bias and noise are modelled as probabilistic functions by specifying distributions over the parameters of a neural network. The basic architecture of a neural network embodying these assumptions is shown in Figure 1. The physical model weights are represented by the output of a softmax layer which enforces the partition of unity constraints and the weighting of model predictions is performed by a subsequent dot product layer. We choose tanh activations for the hidden layer because its mean output is zero-centered, simplifying our prior design. The extrapolation behaviour of a Bayesian Neural Network with tanh activations outside the training set is also predictably flat, which means the predicted model bias and aleatoric noise does not assume unrealistic values even when we extrapolate.

### 2.2 Prior design

The computational expense of Bayesian inference is only justified if we are able to encode our domain knowledge into the prior. This can be a challenge since our function space intuitions about the modelled quantity require translation to distributions over parameter values or architecture choices. Input warping is an essential first step if we want to restrict our prior function space to physically realistic functions [33]. While only three numbers – latitude, longitude and time – suffice to uniquely identify a datapoint, using these directly as inputs would be problematic because the physical model weights, biases and noise generated by such a prior network would be discontinuous across the 180th meridian and would not respect seasonality. In our case study, locations are represented by their Euclidean coordinates $(u, v, w)$ and the time variable is warped onto the 3D helix

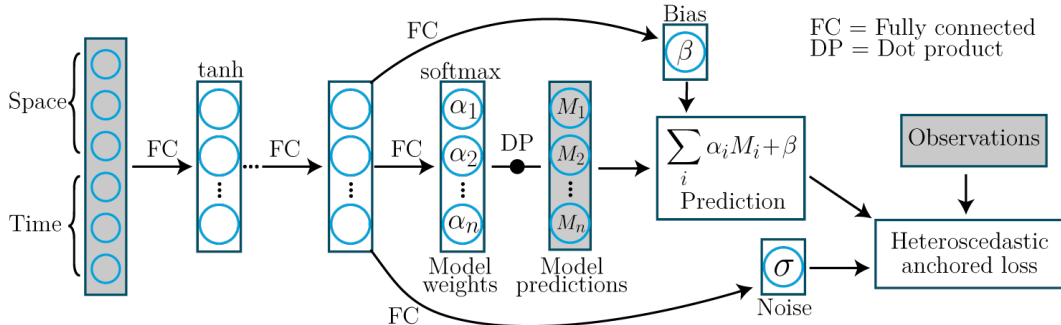

Figure 1: Architecture of the neural network where shading depicts external inputs or data.

$(\cos(2\pi t/T), \sin(2\pi t/T), t)$, where $T = 1$ year. This transformation of the time variable makes our prior network generate weights and biases with both a strong annual periodicity and a slow variation over the years (Figure 2a), consistent with our expectations. The input variables also need appropriate scaling to ensure that our neural network outputs have the desired characteristic lengthscale. Model skills and biases are likely to vary over the typical lengthscales spanned by climatic or geographic regions that and the scaling of the spatial coordinates should be such that this is reflected in the prior. Scaling is also important for temporal input variables and it can be used to magnify or suppress seasonal or yearly variations. Figure 2c demonstrates the effect of three different scaling choices for the $w$ coordinate, $[-0.1, 0.1]$, $[-1, 1]$ and $[-4, 4]$ on randomly drawn samples from the prior: increasing the scale factor favours functions with larger high frequency components.

Another key component of a well-formulated prior is the variance of the pre-softmax layer. Since our physical models are competitive, a priori we should assume that any model combination should be possible at any point in time and space and the $\alpha_i$-s generated by the prior network should be distributed approximately uniformly in the physical model weight simplex. We should therefore choose a prior variance for the incoming connections to the pre-softmax layer such that the variance of the untrained layer outputs is close to 1.0. Figure 2b shows random samples of $\alpha_i$ at an arbitrary point from an untrained neural network with 3 models. It helps us visualize how a small pre-softmax layer standard deviation (0.1) constrains the prior $\alpha_i$ to lie close to the naive multi-model mean, whereas a standard deviation that is too large (4.0) pushes all the prior probability mass towards the corners of the simplex.

Similarly, the prior variances of incoming connections to the model bias term should be scaled to restrict the bias term to zero prior mean and a small variance. While the model bias is necessitated by the fact that certain regions can be modelled poorly by all the physical models, we would prefer to have our combination of physical models do the bulk of the modelling. The prior variances of connections to the heteroscedastic noise term should likewise be scaled. However, unlike the bias term whose distribution should be zero-centered, the noise should have a positive mean added to it that is informed by our knowledge of the average quality of our observations.

Finally, the number of units in the hidden layer(s) should increase commensurately with the size and resolution of our dataset. Overparameterization leads to desirable Gaussian process-like behaviour [23] whereas an underparametrized Bayesian neural network, unable to produce the intricate functions demanded by a large dataset, will have its posterior collapse to a single point and lose all epistemic uncertainty outside of the training dataset. Judicious use of prior predictive checks in conjugation with domain knowledge can thus circumvent the need for expensive hyperparameter tuning or hyperpriors entirely for our simple network and create a well-informed prior.

## 2.3 Approximate inference using randomized MAP sampling

Inference is complicated by the size of a typical geospatial dataset for example, the ozone column dataset used as our case study contains over 2 million datapoints. This rules out more expensive gold-standard inference techniques such as Markov Chain Monte Carlo [29] or Hamiltonian Monte Carlo [7]. Mean-field variational inference [5] or Monte Carlo dropout [14], while scalable, are also inappropriate because one of the objectives of ensembling models is to fill-in missing data and it has been demonstrated [13] that these techniques are excessively overconfident between well-seperated

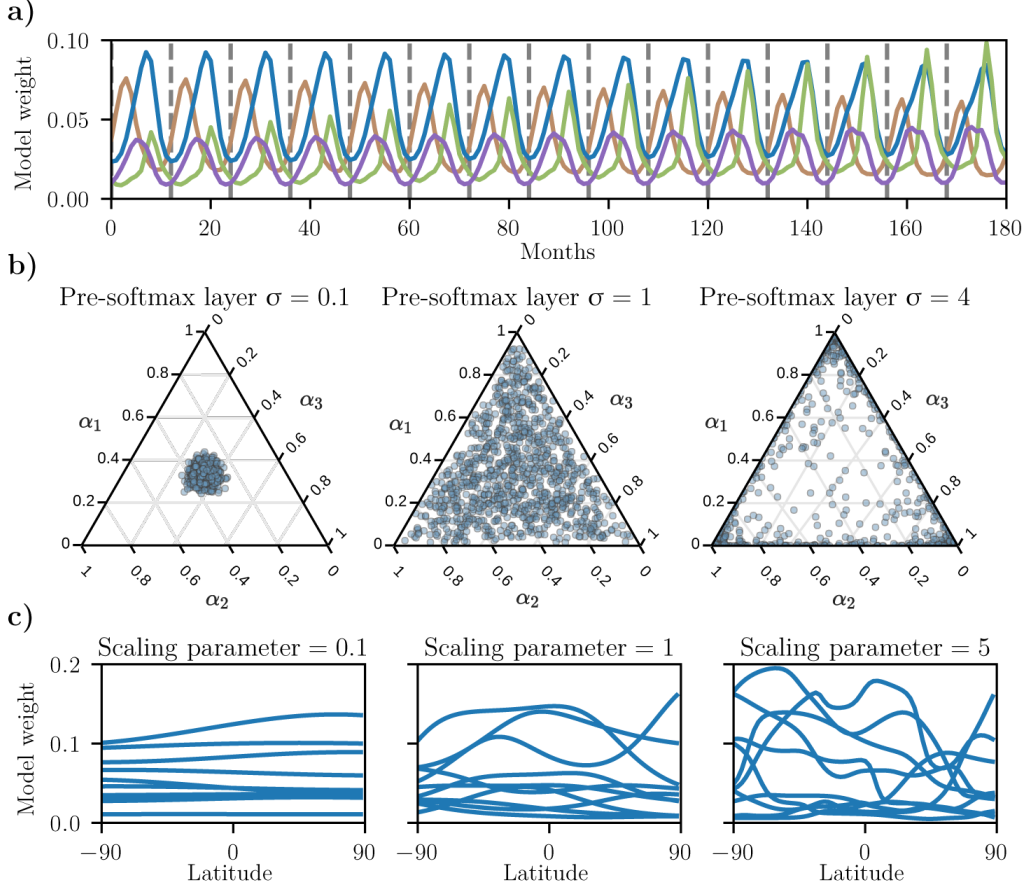

Figure 2: Visualizing the prior. a) shows how the warping of the time coordinate enforces desired quasiperiodicity in the physical model weights generated by the prior. Different coloured lines depict the model weights for four samples from the untrained prior and grey dashed lines split the plot into 12 month intervals. b) shows the effect of pre-softmax layer output variance on the prior distribution of physical model weights in the 2-simplex at a particular point in time and space, using 1000 samples from the neural network prior with 3 physical models. c) shows the impact of scaling spatial input on the lengthscales of physical model weights produced by the neural network prior.

clusters of training data. Here, we have chosen a state-of-the-art approximate inference technique: approximately Bayesian ensembling using randomized maximum a posteriori (MAP) sampling [34]. Ensembling had already been shown [21] to be empirically effective at providing calibrated estimates of uncertainty for neural networks and the randomized MAP sampling approach grounds this in Bayesian theory. For the $j$-th neural network ensemble member, we draw a sample from the prior distribution over parameters (assumed multivariate normal) $\boldsymbol{\theta}_{anc,j} \sim \mathcal{N}(\boldsymbol{\mu}_{prior}, \boldsymbol{\Sigma}_{prior})$ and compute the MAP estimate corresponding to a prior re-centered at $\boldsymbol{\theta}_{anc,j}$.

$$\boldsymbol{\theta}_{MAP,j} = \text{argmax}_{\boldsymbol{\theta}_j} \log(P_{\mathcal{D}}(\mathcal{D}|\boldsymbol{\theta}_j)) - \frac{1}{2}\|\boldsymbol{\Sigma}_{prior}^{-1/2}(\boldsymbol{\theta}_j - \boldsymbol{\theta}_{anc,j})\|_2^2 \qquad (2)$$

If we now consider a dataset of $N_D$ observations $\{y_i, \mathbf{x}_i, t_i\}$ and specify the data likelihood $P_{\mathcal{D}}(\mathcal{D}|\boldsymbol{\theta}_j)$ for our regression task by assuming heteroscedastic Gaussian noise $\sigma(\mathbf{x}, t)$, we may equivalently minimize the following loss function for the $j$-th neural network

$$\text{Loss}_j = \sum_{i=1}^{N_D} \frac{(y_i - \hat{y}_j(\mathbf{x}_i, t_i))^2}{\sigma_j^2(\mathbf{x}_i, t_i)} + \sum_{i=1}^{N_D} \log(\sigma_j^2(\mathbf{x}_i, t_i)) + \|\boldsymbol{\Sigma}_{prior}^{-1/2}(\boldsymbol{\theta}_j - \boldsymbol{\theta}_{anc,j})\|_2^2 \qquad (3)$$

The prediction of a trained ensemble with $n_e$ neural networks is therefore a mixture of $n_e$ Gaussians, each centered at $\hat{y}_j(\mathbf{x}_i, t_i)$ with a variance of $\sigma_j^2(\mathbf{x}_i, t_i)$. For computational convenience, we approximate this mixture as a single Gaussian with mean $\frac{1}{n_e}\sum_j \hat{y}_j$ and variance

$\frac{1}{n_e} \sum_j \sigma_j^2 + \frac{1}{n_e} \sum_j \hat{y}_j^2 - (\frac{1}{n_e} \sum_j \hat{y}_j)^2$, following similar treatment in [21]. This also allows us to decompose the total predictive uncertainty into an aleatoric component (first term) and an epistemic component (second and third terms).

To disambiguate any reference to ensembling or ensembles in this paper, we refer to the combination of geophysical models as the "physical model ensemble" and to the set of neural networks used for approximate inference as the "neural network ensemble".

An outline of the algorithm used to train our BayNNE is provided below.

---

**Algorithm 1:** Algorithm for initialising and training the BayNNE

---

**Input** : Training dataset of $N_D$ observations and physical model predictions corresponding to locations and times $\{\mathbf{x}_i, t_i\}$, physical model predictions for $N_T$ locations and times with missing observations $\{\mathbf{x}_k, t_k\}$

**Output** : Mean and variance predicted by metamodel for $\{\mathbf{x}_k, t_k\}$

1 Transform latitude, longitude and time of each datapoint to 6-dimensional space-time input $(\cos(lat_i)\sin(lon_i), \cos(lat_i)\cos(lon_i), \sin(lat_i), \cos(2\pi t_i/T), \sin(2\pi t_i/T), t_i)$.

2 Rescale each column of space-time inputs to the range $[-a, a]$. Use larger scales $a$ for input variables on which we expect model weights/ bias to have stronger dependence.

3 Set prior variances of the fully connected layer weights to $l_i/n_{input}$, where $n_{input}$ is the number of nodes in the previous layer.

4 Tune $l_i$ by performing prior pushforward checks – the output of each fully connected layer should have mean $\sim 0$ and variance $\sim 1.0$, except those that feed the bias and noise terms, whose output variance should be small.

5 Initialize $n_e$ neural networks by drawing samples from the prior over parameters.

6 **for** $j \leftarrow 1$ **to** $n_e$ **do**

7      Draw a random sample $\boldsymbol{\theta}_{anc,j}$ from the prior over parameters.

8      Anchor the loss function of j-th neural network to $\boldsymbol{\theta}_{anc,j}$, so that

$$\text{Loss}_j = \sum_{i=1}^{N_D} \frac{(y_i - \hat{y}_j(\mathbf{x}_i, t_i))^2}{\sigma_j^2(\mathbf{x}_i, t_i)} + \sum_{i=1}^{N_D} \log(\sigma_j^2(\mathbf{x}_i, t_i)) + \|\boldsymbol{\Sigma}_{prior}^{-1/2}(\boldsymbol{\theta}_j - \boldsymbol{\theta}_{anc,j})\|_2^2.$$

9      Train with ADAM until convergence.

10 **end**

11 **for** $k \leftarrow 0$ **to** $N_T$ **do**

12      $\mu_{pred,k} = \frac{1}{n_e} \sum_j \hat{y}_j(\mathbf{x}_k, t_k)$

13      $\sigma_{pred,k} = \frac{1}{n_e} \sum_j \sigma_j^2(\mathbf{x}_k, t_k) + \frac{1}{n_e} \sum_j \hat{y}_j^2(\mathbf{x}_k, t_k) - (\frac{1}{n_e} \sum_j \hat{y}_j(\mathbf{x}_k, t_k))^2$

14 **end**

15 Compute negative log-likelihood of predictions on test data. If NLL not converged, return to step 5 and train more neural networks.

16 **return** $\mu_{pred}, \sigma_{pred}$

---

## 3 Experiments

### 3.1 Synthetic data

To validate the Bayesian neural network ensembler (BayNNE), we create a toy problem where the ground truth is known. The "monthly observations" are generated by the function $0.5 \left(\frac{\text{lat}}{90}\right)^2 + 0.25 \sin\left(2\pi \frac{\text{lon}}{180}\right) - 0.2 \cos\left(\pi \frac{\text{mon}}{12}\right)$ (lat, lon and mon are latitude, longitude, and month number respectively) with varying levels of added Gaussian noise in different regions to simulate heteroscedasticity– the noise standard deviation is 0.01 in the northern region (north of 30°N), 0.02 in the tropics (between 30°S and 30°N) and 0.03 in the southern region (south of 30°S). The four "physical models" replicate the observations but only in distinct geographical regions: model 1 is correct in the northern region where it has a bias of $+0.03$ w.r.t. the observations, models 2 and 3 are correct and unbiased in the equatorial region and model 4 is correct in the south with a bias of $-0.03$. In regions where the models are not designed to be skilful, they output random noise. Model predictions and observations are shown in Figure 3. The synthetic observations span 20 years, and we train the BayNNE on 85% of the data from the first 10 years. The last 10 years are left for out-of-sample validation.

Results of a BayNNE with 50 neural network ensemble members with 1 hidden layer of 100 nodes trained on this synthetic dataset are shown in Figure 3. We observe that it has successfully recovered the expected physical model weights: models only have weights in the regions where they are skilful and where multiple models are equally skilful (models 2 and 3 in the equatorial region), they are assigned equal weights on average. We also find that the magnitudes of the recovered model biases and aleatoric noise match their engineered values. The uncertainty quantification is excellent out of sample with 68.2, 95.4 and 99.7 percent of points lying within 1, 2 and 3 standard deviations respectively. The overall predictive skill is consistent across the training, testing and out of sample datasets, with near-optimal RMSEs of 0.022, close to the average noise in our observations. This test validates the ability of the BayNNE to successfully capture model skill, bias and aleatoric noise, demonstrating competence in accurate ensembling.

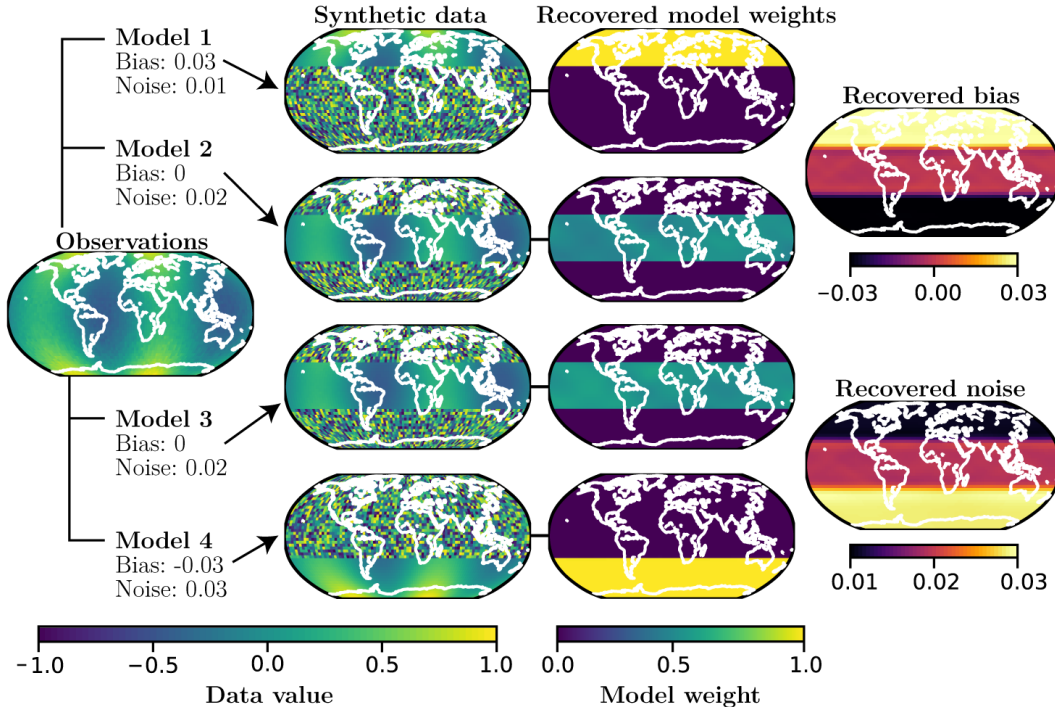

Figure 3: Summary of the toy problem results, showing the synthetic observations and model predictions for a single month 5 years out of sample. These are shown alongside the model weights recovered by the BayNNE for each model, the model bias, and aleatoric noise.

## 3.2 Total column ozone dataset

For a more compelling case study with real-world implications, we consider the problem of predicting monthly averaged total column ozone, which is the integrated amount of ozone from the surface to the atmosphere's boundary with space. Total column ozone provides a good estimate of stratospheric ozone and its variability, as approximately 90% of ozone resides in the stratosphere. Studying and predicting ozone concentrations is an important scientific endeavour, particularly for monitoring the impacts of the Montreal protocol which was designed to protect the ozone layer from anthropogenic emissions [39]. This sustained interest has produced a good coverage of observational records and models suited to simulating ozone, both of which we use.

### 3.2.1 Description of dataset

We use total column ozone output from 15 chemistry-climate models within the Chemistry-Climate Modelling Initiative (CCMI)[28]. This ensemble of process models simulates the changing climate, the chemical composition of the atmosphere, and couplings within the chemistry-climate system. We use the hindcast simulation (1980–2010) from CCMI where the models have been nudged [32] so that they replicate the observed meteorology of the past. This simulation represents the models'

best attempt at recreating the chemical composition for this 30-year period, making it a suitable comparison to observations.

The observations we use are the NIWA-BS total column ozone record [6] which is a dataset constructed from satellites and ground-based observations. Coverage is limited by satellite availability and the method of observation, which for some instruments requires daylight. For this reason, a large proportion of missing observations are over polar regions during winter months, an area of interest as this covers the formation of the ozone hole.

### 3.2.2 Constructing the validation set

Significant spatial and temporal correlations present in geospatial data mean that a randomised train-test split will not adequately validate the skill of the BayNNE. Instead, we withold a set of data for validation which mimics the spatial structure and the temporal occurrence of the data missing from the observations. This is a more rigorous test of the intended application of the BayNNE. To test interpolation, we consider 2 forms of data voids in the observations: data missing from tropical regions (often gaps between satellite tracks) and small irregular missing features. Using historic satellite data, we create synthetic patterns resembling those usually associated with missing data due to incomplete satellite coverage, sometimes covering the majority of the tropics. The total data withheld for the purposes of interpolation validation is 24 months of the entire region $30°$S to $30°$N, 48 months of synthetic data voids due to incomplete satellite coverage and an additional 500 randomly distributed small scale features (up to $15° \times 15°$). We test temporal extrapolation by withholding the last 3 years of data, and spatial extrapolation by withholding data from polar regions. The latter is either an area extending from a latitude of either $60°$ or $70°$ to the pole, which replicates the inability of some instruments to measure ozone at high latitudes during wintertime [17]. In total, 24 months of polar cap data for both the north pole and south pole is used for the purpose of validation. Overall, the BayNNE is trained on 77% (1.8 million datapoints), tested on 4% (85,000 datapoints) and validated on 19% (440,000 datapoints) of the available data.

### 3.2.3 Results

The BayNNE used to ensemble the 15 chemistry-climate models for predicting total column ozone comprises of 65 neural network ensemble members, each containing a single hidden layer with 500 nodes. Comparisons between BayNNE and commonly used ensembling and interpolation methods are shown in Table 1. Interpolation in non-polar regions, including predominantly large gaps in the tropics from incomplete satellite coverage, is compared against bilinear interpolation [1, 25], and spatiotemporal kriging[38, 41] using a stochastic variational Gaussian process [8] on 3 year sections of observational data. Spatial and temporal extrapolation skill (root mean squared error) is compared to a uniformly globally weighted multi-model mean [9, 22] and 2 weighted means where weights per model are found from the ability of a model to replicate observations in the training set [20, 36]. The reader is referred to the supplementary information file for more details on prior design for the BayNNE, training and baseline comparisons.

The BayNNE predictions are significantly better than the baselines in nearly all subsets of the validation dataset (Table 1). Particular improvement over existing methods is seen for ozone predictions over the southern polar cap and for future predictions. Chemistry-climate models are typically less good at simulating ozone over the south pole compared to the north pole due to cold biases [10] and discrepancies in simulating the polar vortex [15, 24]. However, by spatiotemporally identifying skilful models, BayNNE is better at predicting southern polar ozone than other ensembling standards in the modelling community. Skill in temporally extrapolating is also much improved, meaning that using BayNNE may provide more accurate future predictions, which is extremely desirable.

Interpretability, a key benefit of using this technique, is highlighted in Figure 4, allowing for identification of seasons and regions in which particular models contribute more to the ensemble prediction. For example, we identify the three chemistry-climate models with the largest contributions to the ensemble prediction: IPSL, MRI and UMUKCA. IPSL contributes highly, particularly in far northern regions but also globally, and UMUKCA is the most dominant model in the tropics. Also of note (not shown), is the CCSRNIES model, whose weight for the southern pole in austral winter is approximately half, overwhelmingly making it the dominant model for predicting the build-up period before the springtime Antarctic ozone hole.

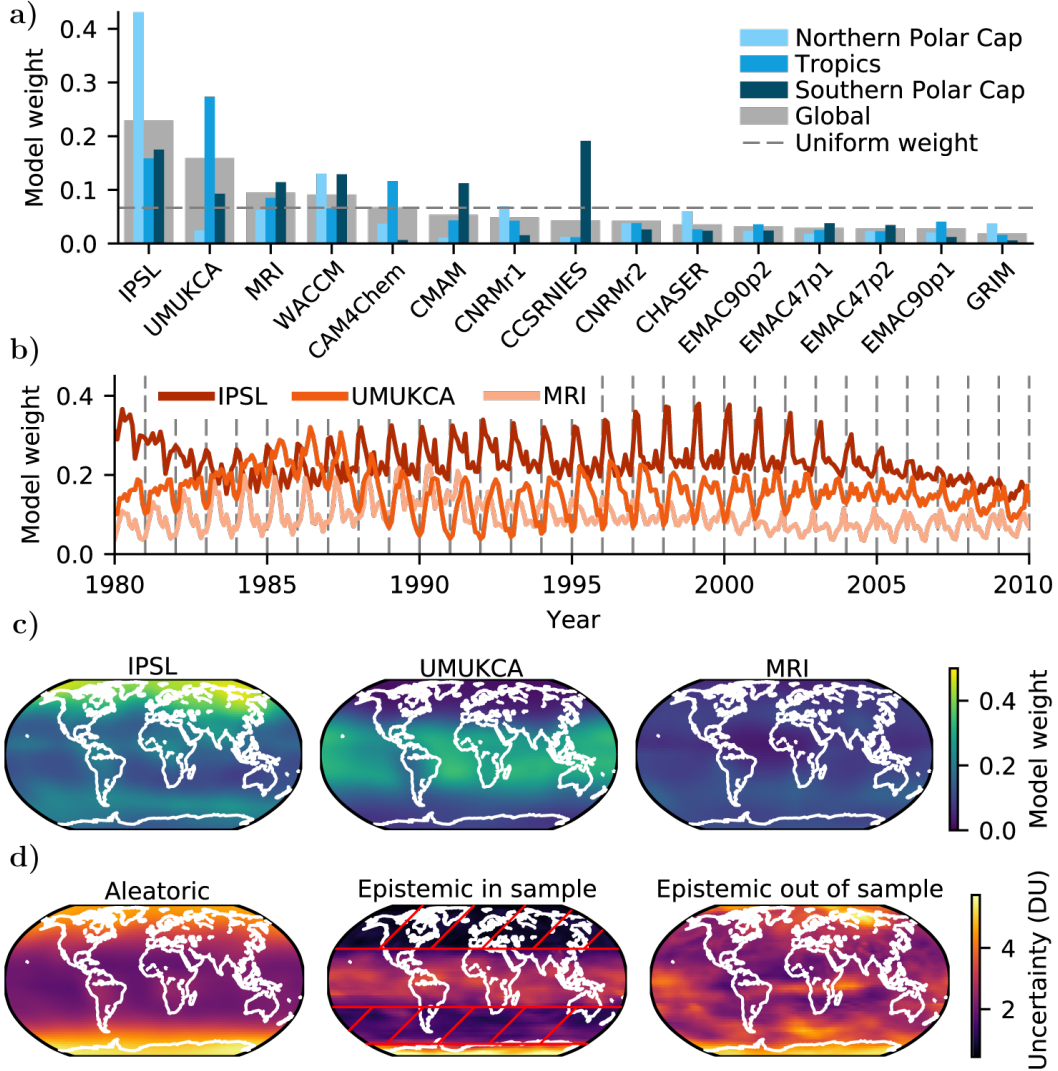

Figure 4: Physical model weights and uncertainties recovered by BayNNE when predicting total column ozone. a) shows the average weight per chemistry-climate model globally and for three regions of interest. b) depicts the spatially averaged model weight for the same three models. Vertical dashed lines show the beginning of the year. c) shows the temporally averaged model weight for the three highest weighted models. d) shows the predicted average aleatoric uncertainty and two temporal snapshots of epistemic uncertainty: in sample (with red hatching depicting areas with available training data), and out of sample. DU is a Dobson unit; a measure of the amount of a trace gas in a vertical column.

Figure 4d shows how the BayNNE successfully handles uncertainty. Epistemic uncertainty is increased for regions lacking observations and temporally out of sample. This is highly desirable behaviour as we do not want an ensembling method to be overly confident beyond data it has seen. To quantify success at handling uncertainty we compare the number of predictions which lie within 1,2 and 3 standard deviations of the truth, which ideally should be 68.3%, 95.5% and 99.7% respectively. These values are: 61%, 92% and 99% for temporal extrapolation; 62%, 94% and 99% for the south pole; and 56%, 87% and 97% for the north pole. The negative log likelihoods for these three regions are $-2.56$, $-2.19$ and $-2.33$.

Table 1: Area weighted root mean squared errors of predictions in Dobson units using various methods. NP and SP represent missing north polar and southern polar cap data respectively. For interpolation 'Tropics' covers a block of missing data 30°S to 30°N, 'Satellite voids' represents the incomplete satellite coverage in the tropics, and small features are up to $15° \times 15°$.

| | Extrapolation | | | Interpolation | | |
|---|---|---|---|---|---|---|
| | Temporal | SP | NP | Tropics | Satellite voids | Small features |
| Multi model mean | 15.7 | 30.5 | 8.8 | 9.8 | 9.2 | 16.4 |
| Weighted mean | 8.7 | 22.1 | 12.3 | 8.2 | 8.5 | 10.2 |
| Spatially weighted mean | 9.8 | 19.6 | 6.6 | 5.5 | 5.2 | 10.0 |
| Spatiotemporal kriging* | – | – | – | 7.0 | 2.2 | 3.4 |
| Bilinear interpolation* | – | – | – | 31.2 | **1.7** | 3.4 |
| BayNNE | **4.4** | **6.6** | **4.7** | **2.7** | 2.1 | **3.2** |

\* Used for interpolation only

## 4 Conclusions

We have presented Bayesian neural network ensembling (BayNNE), a principled approach to geophysical model ensembling, which learns spatiotemporally varying model weights and bias based upon the physical models' ability to replicate observations. This uncertainty-aware approach incorporates a heteroscedastic aleatoric uncertainty, which accounts for the varying quality of observational data and other sources of irreducible uncertainties. Additionally, the epistemic uncertainties inherent in a Bayesian framework prevent overconfident extrapolation when BayNNE is not constrained by observations.

We have validated BayNNE on a synthetic dataset where we demonstrated its ability to recover the correct model weights, biases and noise. We then applied it to the more challenging problem of ensembling 15 chemistry-climate models to predict total column ozone. BayNNE predictions were significantly more accurate than current ensembling techniques, both temporally out-of-sample and for infilling historic observations. As a result, we have produced an accurate and complete gridded reconstruction of total column ozone for the period 1980–2010, which offers new insights, particularly for the ozone hole. Interpretability is maintained and model weights/ biases offer an understanding of localised model performance, allowing diagnosis by modellers. Considering that most physical model ensemble weighting techniques do not vary weights in space and time and do not take account of uncertainties in the observations used to create model weights, this ensembling technique represents a significant improvement.

Accurately ensembling geophysical models (e.g. climate models) improves the predictive capability of the ensemble, allows for better investigation of historic conditions through the imputation of discontinuous observations and in the case of climate models, is vital for investigating the evolution of the climate. Moreover, quantifying the certainty of predictions is fundamental for constraining future change and describing our confidence in the predictions. Future work should not only look at applying this tool to other climate modelling problems but also to problems in other disciplines, such as hydrology, where competing model predictions need to be similarly combined in light of observational evidence. We note that the nudged chemistry-climate models in our case study have their behaviour partially constrained by observed meteorology, whereas free running models predicting the future cannot have this constraint. A proper treatment of the chaos-induced uncertainty in free running models would be worth investigating for use in forecasting using ensembles. Finally, it would also be interesting to consider physical model weights as a function of model variables (e.g. ozone-temperature gradient) that causally impact model skill, instead of proxies like location and time, as this may improve forecast accuracy.

## Broader Impact

We created an ensembling technique which takes into account the limitations of observations and models. This method is applicable to many geophysical models (e.g. hydrological, regional climate and chemistry-climate models) though nuances in each field and model ensemble mean the BayNNE should not be blindly used.

Positive impacts include more accurate and better constrained predictions from model ensembles. This could shift the standard of how model ensembling is performed, leading to this method (or derivatives) influencing scientific understanding and downstream policy decisions. The greater understanding offered by combing models and observations in this way, has the potential to open up sparse historic observational records, through fusion with geophysical models. This would, for example, allow for greater understanding of historic climate states.

The response to climate change is influenced by predictions formed from models ensembles, and though accurate and appropriately certain ensembling could result in more definitive and correctly concentrated mitigation efforts, highly certain but wrong predictions could lead to an incorrect pooling of resources and result in negative socio-economic impacts. For these reasons we must be mindful about dangers of extrapolating and unknown errors in observational datasets which incorrectly bias results.

## Acknowledgments and Disclosure of Funding

This work was supported by the Natural Environment Research Council [NERC grant reference number NE/L002604/1], with Matt Amos's studentship through the ENVISION Doctoral Training Partnership. Ushnish Sengupta is an Early Stage Researcher within the MAGISTER consortium which receives funding from the European Union's Horizon 2020 research and innovation programme under the Marie Skłodowska-Curie grant agreement No 766264. The project was also supported with research credits provided by Google Cloud.

Paul J Young is partially supported by the Data Science of the Natural Environment (DSNE) project, funded by the UK Engineering and Physical Sciences Research Council [EPSRC grant number EP/R01860X/1].

We acknowledge the modelling groups for making their simulations available for this analysis, the joint WCRP SPARC/IGAC Chemistry-Climate Model Initiative (CCMI) for organising and coordinating the model data analysis activity, and the British Atmospheric Data Centre (BADC) for collecting and archiving the CCMI model output. We would like to thank Bodeker Scientific, funded by the New Zealand Deep South National Science Challenge, for providing the combined NIWA-BS total column ozone database.

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
