[Supplementary Material]

# Supplementary material: Ensembling geophysical models with Bayesian Neural Networks

# Contents

## A  Construction of ozone baselines

We remind the reader that all of these baselines use the same data for training, testing and validation as the Bayesian neural network ensemble. This validation tests the ability of the ensembling methods to interpolate and extrapolate, particularly over regions of interest and sparse data.

### A.1  Multi-model mean

This is the uniform weighting of all the 15 chemistry-climate models. The prediction is therefore,

$$y_{\text{MMM}}(\mathbf{x}, t) = \frac{1}{15} \sum_{i=1}^{15} M_i(\mathbf{x}, t) \qquad (1)$$

where $y_{\text{MMM}}(\mathbf{x}, t)$ is the multi-model mean prediction and $M_i(\mathbf{x}, t)$ is the i-th individual model prediction.

## A.2 Weighted mean

Here the ensemble mean is constructed from model projections weighted by their skill (in replicating observations) and their independence. This is based on work from Knutti et al. [2], Sanderson et al. [3].For an ensemble of $N$ models, the weight for model $i$ ($w_i$) is given by

$$w_i = \exp\left(-\frac{D_i^2}{n_i\sigma_D^2}\right) \Big/ \left(1 + \sum_{j\neq i}^{N} \exp\left(-\frac{S_{ij}^2}{n_i\sigma_S^2}\right)\right), \tag{2}$$

where $D_i^2$ is the squared difference between a model and observation averaged over all space and time, $S_{ij}^2$ is the squared difference between models averaged over all space and time, $n_i$ is the size of data used in constructing the weight, and $\sigma_D$ and $\sigma_S$ are constants which allow tuning of the weighting. The weights $w_i$ are normalise to sum to 1. The weighted prediction is therefore

$$y_{\text{WM}}(\mathbf{x}, t) = \sum_{i=1}^{N} w_i M_i(\mathbf{x}, t). \tag{3}$$

Values of $\sigma_D$ and $\sigma_S$ were found by minimising the difference between the weighted prediction and the observations over the training data.

## A.3 Spatially weighted mean

The ensemble is constructed much the same as the weighted mean presented above, except that model weights vary in space. The weights are calculated

$$w_i(\mathbf{x}, t) = \exp\left(-\frac{D_i(\mathbf{x}, t)^2}{n_i(\mathbf{x}, t)\sigma_D^2}\right) \Big/ \left(1 + \sum_{j\neq i}^{N} \exp\left(-\frac{S_{ij}(\mathbf{x}, t)^2}{n_i(\mathbf{x}, t)\sigma_S^2}\right)\right), \tag{4}$$

and are used to generate the prediction,

$$y_{\text{SWM}}(\mathbf{x}, t) = \sum_{i=1}^{N} w_i(\mathbf{x}, t) M_i(\mathbf{x}, t). \tag{5}$$

## A.4 Spatiotemporal kriging

We performed spatiotemporal kriging using an implementation of a stochastic variational Gaussian process (SVGP) from GPFlow [1]. Due to the size of the training data (1.8 million data points) we used a SVGP on 3 year sections of observational data with 5000 inducing points per section. The kernel we used was the product of a Matern3/2 kernel acting over latitude and time, and periodic Matern3/2 kernel acting over longitude. We used an Adam optimiser implemented in tensorflow to train the SVGP.

## A.5 Bilinear interpolation

Bilinear interpolation over the training data was performed using the SciPy function grid-data: `https://docs.scipy.org/doc/scipy/reference/generated/scipy.interpolate.griddata.html`.

# B  Hyperparameter details

The pretrained model weights and the code to run the BayNNE for both the synthetic and ozone experiments can be found here: `https://anonymous.4open.science/r/6bf08e5a-c909-45a3-be63-aa0f5ba187df/`. Table 1 shows the hyperparameters chosen in the running of the BayNNE for both experiments.

The heteroscedastic loss function is prone to episodes of catastrophic forgetting. To avoid these, we use large batch sizes, small learning rates and a large number of epochs so that each neural network ensemble member may be stably trained till convergence.

Table 1: Hyperparameter values and priors for BayNNE.

| | Synthetic experiment | Ozone experiment |
|---|---|---|
| Spatial coord scaling | 2 | 2 |
| Temporal coord scaling (month of year) | 1 | 2 |
| Temporal coord scaling (total months) | 1 | 1 |
| Number of physical models | 4 | 15 |
| Number of neural network ensemble members | 50 | 65 |
| Bias mean. prior | 0 | 0 |
| Bias std. prior | 0.01 | 0.03 |
| Noise mean prior | 0.02 | 0.015 |
| Noise std. prior | 0.004 | 0.003 |
| Number of hidden layers | 1 | 1 |
| Number of hidden nodes | 100 | 500 |
| Optimiser | Adam | Adam |
| Batch Size | 2000 | 25000 |
| Learning rate | $5 \times 10^{-5}$ | $3 \times 10^{-5}$ |
| Number of epochs | 6000 | 125000 |

The neural network ensemble for the 2 million datapoint ozone dataset were trained across a cluster of 6 P100 GPUs. Each neural network needed 20 hours to be trained till convergence and the entire ensemble needed 8 days of wall clock time.

# C Derivation of loss function

In the following, we derive the anchored ensembling loss function for the heteroscedastic case. For the $j$-th neural network ensemble member in randomized MAP sampling, we compute the MAP estimate corresponding to a recentered prior over parameters $\boldsymbol{\theta}_{anc,j}$, $P(\boldsymbol{\theta}_j) = \mathcal{N}(\boldsymbol{\theta}_{anc,j}, \boldsymbol{\Sigma}_{prior})$. Here $\boldsymbol{\theta}_{anc,j}$ is a sample from the original multivariate normal prior over parameters, i.e. $\boldsymbol{\theta}_{anc,j} \sim \mathcal{N}(\boldsymbol{\mu}_{prior}, \boldsymbol{\Sigma}_{prior})$.

$$
\begin{aligned}
\boldsymbol{\theta}_{MAP,j} &= \operatorname{argmax}_{\boldsymbol{\theta}_j} P(\boldsymbol{\theta}_j|\mathcal{D}) \\
&= \operatorname{argmax}_{\boldsymbol{\theta}_j} P_{\mathcal{D}}(\mathcal{D}|\boldsymbol{\theta}_j)P(\boldsymbol{\theta}_j) \ \text{ (Bayes' Theorem)} \\
&= \operatorname{argmax}_{\boldsymbol{\theta}_j} \log(P_{\mathcal{D}}(\mathcal{D}|\boldsymbol{\theta}_j)) + \log(P(\boldsymbol{\theta}_j)) \ \text{ (log strictly monotonic increasing)} \\
&= \operatorname{argmax}_{\boldsymbol{\theta}_j} \log(P_{\mathcal{D}}(\mathcal{D}|\boldsymbol{\theta}_j)) - \frac{1}{2}(\boldsymbol{\theta}_j - \boldsymbol{\theta}_{anc,j})^T \boldsymbol{\Sigma}_{prior}^{-1}(\boldsymbol{\theta}_j - \boldsymbol{\theta}_{anc,j}) + \text{const.} \\
&= \operatorname{argmax}_{\boldsymbol{\theta}_j} \log(P_{\mathcal{D}}(\mathcal{D}|\boldsymbol{\theta}_j)) - \frac{1}{2}\|\boldsymbol{\Sigma}_{prior}^{-1/2}(\boldsymbol{\theta}_j - \boldsymbol{\theta}_{anc,j})\|_2^2 \ \text{ (diagonal prior covariance)}
\end{aligned}
$$

If we specify the data likelihood for our regression task assuming i.i.d. observations and additive heteroscedastic Gaussian noise i.e., $P_{\mathcal{D}}(\mathcal{D}|\boldsymbol{\theta}_j) = \prod_{i=1}^{N_D} \mathcal{N}(\hat{y}_j(\mathbf{x}_i, t_i), \sigma_j^2(\mathbf{x}_i, t_i))$, we obtain

$$
\begin{aligned}
\boldsymbol{\theta}_{MAP,j} &= \operatorname{argmax}_{\boldsymbol{\theta}_j} -\frac{1}{2}\sum_{i=1}^{N_D} \frac{(y_i - \hat{y}_j(\mathbf{x}_i, t_i))^2}{\sigma_j^2(\mathbf{x}_i, t_i)} - \sum_{i=1}^{N_D} \log(\sigma_j(\mathbf{x}_i, t_i)) + \text{const.} - \frac{1}{2}\|\boldsymbol{\Sigma}_{prior}^{-1/2}(\boldsymbol{\theta}_j - \boldsymbol{\theta}_{anc,j})\|_2^2 \\
&= \operatorname{argmin}_{\boldsymbol{\theta}_j} \sum_{i=1}^{N_D} \frac{(y_i - \hat{y}_j(\mathbf{x}_i, t_i))^2}{\sigma_j^2(\mathbf{x}_i, t_i)} + \sum_{i=1}^{N_D} \log(\sigma_j^2(\mathbf{x}_i, t_i)) + \|\boldsymbol{\Sigma}_{prior}^{-1/2}(\boldsymbol{\theta}_j - \boldsymbol{\theta}_{anc,j})\|_2^2 \ \text{ (}\times \text{ -2 throughout)}
\end{aligned}
$$

# D  Extra ozone experiment plots

In the main text we highlighted the models with the most interesting features and highest weights. For completeness here, we include a wider range of plots looking at model weights and modelled bias and uncertainties, for the ozone experiment.

## D.1  Bias

Figures 1 and 2 show the modelled bias averaged in time and space respectively. Bias is seen to be negative over polar regions especially the southern polar region and southern mid latitudes.

Figure 1: Temporally averaged modelled bias.

Figure 2: Spatially averaged modelled bias.

## D.2  Epistemic uncertainty

Figures 3 and 4 show the epistemic uncertainty averaged in time and space respectively. Epistemic uncertainty is highest at polar regions. Epistemic uncertainty increases for regions with sparse or no data including 2007–2010 (used to validate extrapolation) and 1993–1997 where there is a greater sparsity of data. This can be seen clearly in Figure 4.

## D.3  Average model weight

Figure 5 shows the average model weight for all 15 chemistry-climate models used in the ozone experiment.

Figure 3: Temporally averaged epistemic uncertainty.

Figure 4: Spatially averaged epistemic uncertainty and the number of training points per month.

Figure 5: Temporally averaged model weights for all 15 chemistry-climate models.