[Reviews · NeurIPS 2020]

Review 1

Summary and Contributions: The paper proposes a method to merge the predictions of an ensemble of models using an ensemble of bayesian neural networks. In particular the method is applied to an ensemble of physical models of the Ozone column in the atmosphere (lon, lat, time dimensions) and how these should be mixed across both space and time. Experimental results are clearly presented for both a sanity checking toy example as well as real world historical data. The paper implements a set of baselines based on published alternative methods and their results compare favourably to these. #### Answer to Rebuttal #### First, thanks for the author response. My concerns about the method section not being completely self-contained have been addressed. Secondly i also found the NLL vs ensemble members plot requested by reviewer 2 relevant to include in the appendix. ####

Strengths: First and foremost I think the paper tackles an important and interesting problem. In general I found the paper very well written and mostly easy to follow (see exemption below). The experiments are sound, interesting and clearly illustrate the benefit of the method compared to more naive approaches. An illustrative toy experiment further clearly showed that the model can recover the ground truth, albeit in a somewhat simple setting.

Weaknesses: My main concerns with the paper is that some of the method sections are have details missing and is not completely self-contained, in particular: It is not clearly explained how Eq (2) is derived - The reader could consult the ref, however this is a key part of the paper and should be explained in a self-contained way. Exactly how do you train the ensemble of Bayesian neural networks using eq (2) and eq (3) including the sampling process of theta_anc, please describe that clearly maybe using pseudo code. Please clearly state what the prior assumptions is of the neural network parameters and how the stochastic MAP solution approximates the posterior over these. Experimental details are missing (many seems to have been relegated to the appendix, please prioritize more of these details in the main paper) What are the ensemble sizes, training times, number of parameters etc Is Sigma_prior a diagonal covariance? Lastly, the method is not particularly new, however I think the experimental results, presentation and interesting application renders this a minor issue. I’ll be happy to bump my score if you describe the method and experimental sections in more details

Correctness: Method, Results and theory all seem sound to me.

Clarity: In general the paper is very well written - however what does the term ‘Skill’ exactly mean in this context?

Relation to Prior Work: Nothing to add

Reproducibility: Yes

Additional Feedback:


Review 2

Summary and Contributions: This paper builds an ensemble of geophysical models using a weighting scheme derived in itself from an ensemble of NNs. The authors use randomised MAP sampling to mimic the result of sampling from the posterior of a BNN. The result is a model that is able to provide interpretable results which includes uncertainty quantification. ############### Response to Rebuttal ############### Thanks very much for the author response and directly responding to my questions. I think the pseudocode and the related work paragraph will make great additions to the paper. In particular, the pseudocode will answer some of the questions both R1 and myself had about sampling. I also am pleased to see the plot of the NLL vs ensemble number. It makes it easier to argue for your choice of the ensemble. It might be worth putting that in the appendix. Finally, perhaps in future work, exploring other BNN approaches might be interesting. Given the author response, I am happy to stay with the current score for the paper.

Strengths: This paper is well-motivated and provides impressive results. The strength of this work is that the model is shown to be interpretable, both in the way the geophysical models are combined and also in how uncertainty is clearly represented. The use of figures in the paper is also of a very high quality and makes it easy for the reader to understand what is being represented.

Weaknesses: My main concerns are the following: 1) Although well-motivated, the paper does not include a related work section to help place this work in the community. Have others worked on combining ML with climate models before? I am aware of many works in Astrophysics that have done so, but I am not too familiar with this domain and would benefit from a summary of others working on similar ML approaches. 2) I am not completely comfortable with the definition of the “z coordinate” (line 79). Is this the prior variance of the weights? 3) Line 103 advocates for “judicious use of prior predictive checks”. I would be interested to hear more details about this and how it was applied here. 4) Did you try other techniques (e.g. MC dropout etc… ) and find they were over-confident in this scenario? 5) How do you define the mean and covariance of \theta_{anc,j}? And do you sample \theta_{anc,j} at training time? What about at test time? 6) How did you do hyperparameter optimisation? Why did you pick 65 NNs in your ensemble, did you try more / fewer and get a different performance? 7) Another baseline that could be used is to pick the best performing geophysical model at {x_i,t}, I would expect your ensemble model to do better but would possibly be interesting. Minor: 1) Line 119: Notation inconsistency -> introduces {x_i, y_i, t_i}, where bold x is used elsewhere. 2) In Figure 4d) acronym DU is not defined

Correctness: Please see above for more details.

Clarity: This is a strength of the paper.

Relation to Prior Work: This is not clear to me from the paper. As mentioned above, the paper would benefit from including other works that have applied ML with geophysical models.

Reproducibility: Yes

Additional Feedback: The authors provide code and supplementary materials. Therefore they are strong in this domain.


Review 3

Summary and Contributions: This manuscript introduces a Bayesian Neural Network approach to modeling geophysical data, which shows promising results in the context of chemistry-climate predictions. The authors here show that by adjusting to model-specific uncertainty the ensembling is significantly improved in extrapolation. I read the author rebuttal and did not change my opinion. As suggested by one of the reviewers, the authors should carefully consider how to disseminate this work to the geophysics field.

Strengths: This mansuscript demonstrates that using a Bayesian neural network in the context of geophysical modeling can improve interpolation and extrapolation. This is shown on real-world data and synthetic data, and especially seems to improve over simpler model averaging techniques. Largely, this is a nice application paper. The input representation of the time and spatial locations is clever. I do not know whether that is standard or not--the authors should clarify whether they consider this a contribution. The theoretical grounding of the algorithm is sound. The empirical evaluation is good, and I appreciate that care went into choosing the validation set. This article will only be relevant to a subset of the NeurIPS community.

Weaknesses: The machine learning methodological contributions here are small. Largely, the models are relatively standard, and the comparisons are largely to simpler models. The ability of the models to estimate uncertainty accurately is not quantitatively evaluated. It would be beneficial to the manuscript to provide metrics on the uncertainty in addition to the pure RMSE performance and the qualitative analysis on the uncertainty.

Correctness: Yes. Care was taken on choosing the validation set as well a clear description. Given that validation in this area is often sloppy, I really appreciated this.

Clarity: Yes, the paper is clear.

Relation to Prior Work: It seems reasonable, but it is not overly discussed.

Reproducibility: Yes

Additional Feedback:


Review 4

Summary and Contributions: The paper proposes using a Bayesian neural network to combine physical models to produce a more accurate prediction. The Bayesian neural network is trained via supervised learning using a measured physical quantity as a target.

Strengths: Provides an accurate way to combine physical model estimates of ozone, with quantified uncertainty.

Weaknesses: Bayesian NN not tested versus other standard regression techniques. Simply combining predictions using regression is an old idea in ML, and hence is not very relevant to NeurIPS.

Correctness: The evaluation of the Bayesian NN versus other regression techniques is somewhat weak. The other supervised learning algorithms chosen were linear interpolation on a grid and an unusual form of GPR. Given that this is a simple regression problem, I would try Random Forest regression (available in scikit-learn) and non-Bayesian neural networks. It's not clear how much the Bayesian NN buys in terms of accuracy. It's also not clear whether the uncertainty of the model is at all calibrated.

Clarity: I was initially quite confused by the presentation of the paper. Bayesian ensembling is often done in the context of unsupervised learning, where the true answer is not known. For example, in classification, the classic paper of this form is (Dawid & Skene, 1979). I was trying to understand the paper in this context, but then I realized by the methods secton that the paper is about supervised learning, where an observed quantity is the target of a regressor. That made the whole paper make much more sense.

Relation to Prior Work: The paper should mention previous work in supervised ensembling. One example is "Stacked Regressors" by Breiman, 1996, and references therein. The math setup of Breiman is not identical to that of this paper, but it is worth referencing, I think.

Reproducibility: Yes

Additional Feedback: From an ML research point of view, this paper is a relatively straightforward application of regression using a Bayesian NN. This would have much more impact at AGU than as a poster in NeurIPS. ----- The author rebuttal did not (and perhaps could not) address my point: this paper (if accepted) will be a poster at NeurIPS and will probably disappear without having much impact. The paper would have far more impact in a geophysics context, like AGU.

[Author Response · NeurIPS 2020]

We thank the reviewers for their insightful and positive comments and the area chair for their consideration. We will address common concerns first before addressing individual reviewer comments.

**1. Methodological details.** The comments on insufficient details in the methods section are well-taken. We will add pseudocode that describes both the prior design and the training processes. **2. Novelty/relevance.** While we concede that this paper does not contain a major methodological leap, we believe it combines different elements (some borrowed from prior works) in a novel manner and lays out a roadmap for handling an important class of problems– noisy geospatial time-series data with competing physical/ empirical models of the underlying process where the task is to make forecasts or fill in large gaps. Each one of these elements is crucial to the success of the method: the pointwise-linear model combination offers immediate interpretability and makes it more palatable to domain specialists; the spatio-temporally dynamic model-weighting allows accurate predictions; the enforced quasi-periodicity in time and periodicity in space makes it extrapolate more successfully; the Bayesian framing lets us compute appropriate confidence intervals; the heteroscedastic treatment of aleatoric uncertainty accounts for the highly variable quality of available data; and the ensembling approach to Bayesian inference allows scalability. Further, given the significance of this class of problems, which includes not just climate modelling but also predicting crop yields, pollutants and disease spread, we believe it deserves the attention of the ML community broadly and not just that of domain specialists. We hope that this work can be built upon by others in the ML community and encourage cross-pollination between fields. **3. Related work.** We will be including a paragraph in the introduction which highlights recent uses of ML with climate ensembles (e.g. E. Barnes, P. Nowack, K.L. Chang). However, we believe that our work is distinctly positioned compared to these, which either lack certain considerations (no spatiotemporal weighting or uncertainty), aren't scalable, or are asking different research questions entirely. This is a young subfield, so there are few highly relevant papers. **4. Baselines.** The baseline methods are simple because these are the ones that are widely used by the community. Pure RMSE performance is not the goal here and any method that hopes to replace current standards must balance interpretability, accuracy and a good treatment of uncertainty. **5. Uncertainty evaluation.** Concerns were expressed that the uncertainty of the model has not been quantitatively evaluated. However, we clarify that in the paper we reported the fraction of data points in different subsets of the validation dataset within 1x/2x/3x of predicted standard deviations, allowing a quantitative comparison with the fractions (68.2%/95.4%/99.6%) expected within those ranges for the assumed Gaussian distribution. This point will be made more clearly and mean negative log likelihoods will also be reported.

**Reviewer 1.** Eq. 3 has been derived in the appendix since the heteroscedastic case was not considered in Pearce et al. The derivation of Eq. 2, however, is relatively lengthy, and not a contribution from us. To avoid duplication of their work, we believe it best to omit it from our paper, although we will add details on the assumptions this inference technique makes about parameters of the network. We also agree that the term "model skill" is ambiguous and unnecessary. We will remove it.

**Reviewer 2.** Responding to their numbered comments. **2)** The 'z coordinate' refers to the spatial z coordinate, which is the cosine of latitude. We agree that $x$ and $y$ are overloaded and will clean up our notation. In line 70, Euclidean spatial coordinates $(x, y, z)$ will be changed to $(u, v, w)$. **3,5)** We will add pseudocode to the methods section to further detail the prior design process and what exactly these checks are. **4)** We did not test MC (Monte Carlo) dropout since the Foong et al. paper cited makes a convincing empirical and theoretical case against it. MC dropout also performs worse compared to RMS in benchmark tests by Pearce et al.

Mean NLL with standard deviation (shaded) from random shuffles of the ensemble.

**6)** Hyperparameter optimization was mostly avoided (please see prior design section in the paper). For choice of ensemble size, please see the above plot which shows how the negative log likelihood of the test data converges as we use a larger neural network ensemble. Any ensemble greater than 30 in size would have been adequate but we ran more to ensure convergence nevertheless. **7)** The best overall model and best model at every spatial location will be added as baselines (they perform predictably worse). **Minor comments:** DU (Dobson Unit) definition will be added.

**Reviewer 3.** The space-time representation, though commonsensical, is not standard, to the best of our knowledge. This treatment of coordinates is crucial in ensuring good accuracy.

**Reviewer 4.** We believe that a stochastic variational Gaussian process is a fair comparison as the more standard GPR scales as $O(N^3)$ in the number of data points and our dataset has 1.5 million datapoints. The SVGP paper (Titsias 2009) has accrued 815 citations on Google Scholar and is certainly one of the most common ways of scaling Gaussian Processes to big data. We will cite Breimann (1996), but we observe that that regression involves a statically weighted combination of regressions. If the models are not weighted differently in different parts of the input space, the performance will be similar to the weighted mean baseline.

[Meta-Review · NeurIPS 2020]

The reviewers converged towards recommending to accept this submission. The reviewers were satisfied with the authors' response, and have updated their reviews accordingly. In my decision I discounted the review of R4 given their low confidence and lack of engagement in the discussion.